# Irrigation Regimes and Nitrogen Rates as the Contributing Factors in Quinoa Yield to Increase Water and Nitrogen Efficiencies

**DOI:** 10.3390/plants11152048

**Published:** 2022-08-05

**Authors:** Maryam Bahrami, Rezvan Talebnejad, Ali Reza Sepaskhah, Didier Bazile

**Affiliations:** 1Water Engineering Department, Shiraz University, Shiraz 7144113131, Iran; 2Drought Research Center, Shiraz University, Shiraz 7144113131, Iran; 3CIRAD, UMR Savoirs, Environnement et Sociétés (SENS), CEDEX, 34398 Montpellier, France; 4SENS, CIRAD, IRD, Université Paul Valery Montpellier 3, Univ Montpellier, CEDEX, 34199 Montpellier, France

**Keywords:** *Chenopodium quinoa* Willd., deficit irrigation, nitrogen fertilizer rate, water use efficiency, nitrogen uptake, residual soil NO_3_-N

## Abstract

Sustainable field crop management has been considered to reach the food security issue due to global warming and water scarcity. The effect of deficit irrigation and nitrogen rates on quinoa yield is a challenging issue in those areas. In this regard, the interaction effects of different N rates (0, 125, 250, and 375 kg N ha^−1^) and irrigation regimes [full irrigation (FI) and deficit irrigation at 0.75 FI and 0.5 FI] on quinoa yield and water and nitrogen efficiencies were evaluated with a two-year field experiment. Increasing nitrogen fertilizer application levels from 250 to 375 kg N ha^−1^ under FI and deficit irrigation did not cause a significant difference in seed yield and the total dry matter of quinoa. Furthermore, 20% and 34% reductions were observed for nitrogen use efficiency (NUE) and nitrogen yield efficiency with the application of 375 kg N ha^−1^ compared with that obtained in 250 kg N ha^−1^ nitrogen fertilizer, respectively. Therefore, a Nitrogen application rate of 250 kg ha^−1^ and applying 0.75 FI is suggested as the optimum rate to reach the highest seed water use efficiency (0.7 kg m^−3^) and NUE (0.28 kg m^−3^) to gain 4.12 Mg ha^−1^ quinoa seed yield. Under non–limited water resource conditions, an FI and N application rate of 375 kg ha^−1^ could be used for higher seed yield; however, under water-deficit regimes, an N application rate of 250 kg ha^−1^ could be adequate. However, questions about which environmental factors impressively restricted the quinoa growth for optimizing the potential yield need further investigation.

## 1. Introduction

Recent studies revealed that crop production must be increased by around 25–70% to meet the worldwide food demands in 2050 [1,2] due to rapid population growth. Therefore, it is expected to use sustainable field management [3] and use resilient crops to increase crop products and reach the food security mission; whereas environmental changes [4] such as global climate change [5] and scarce water availability have put negative pressure on agricultural production [6,7,8].

Quinoa crop (*Chenopodium quinoa* Willd.) is an alternative option for farmers due to its adaptability to various agroecosystems [6,9,10], which has been considered widely in the last decades [7,9,10,11]. The adaptation of quinoa to some environmental changes such as drought or water stress [12,13], salinity [14,15,16], and frost [17] has made it favorable amongst scientists for securing food production [18,19,20,21]. A systematic review showed that the mean value of quinoa seed yield amongst the Asian countries (Iran, Israel, Pakistan, Turkey, United Arab Emirates) was about 2.55 Mg ha^−1^ (in the range of 0.01–6.35 Mg ha^−1^) while it was about 2.65 Mg ha^−1^ in South America (ranging from 0.55 to 7.8 Mg ha^−1^). Africa, Europe, and North America (with the lowest mean seed yield, 0.65 Mg ha^−1^), respectively, were placed after them [16]. Quinoa seed is rich in proteins, fats, fiber, minerals, vitamins, and essential amino acids [22,23,24], which extensively enhanced quinoa’s popularity.

Deficit irrigation has been introduced as a proper irrigation regime to increase water use efficiency in quinoa production [14,25], which has been an acceptable strategy, especially in arid and semi-arid areas that suffer from water scarcity [9,26]. Therefore, it could be a valuable option to reach the optimum quinoa seed yield in areas with water scarcity [27,28,29,30]. In this regard, findings by Geerts et al. [25] based on some field experiments in Bolivia and crop modeling, indicated that a deficit in irrigation to 55% of full irrigation had no significant effect on dry matter, while it enhanced seed yield and water use efficiency [25]. Nevertheless, another study in Bolivia reported the flowering and seed filling stages as the most sensitive stages to the water stress conditions [31]. It is correlated with exposure to high temperatures at this stage that have cumulative effects [6,32,33]. Therefore, deficit irrigation and high temperatures at the flowering and seed filling stages play an important role in quinoa seed yield.

Furthermore, nitrogen plays a vital role in increasing crop productivity as it increases crop yield for the unit of applied water [34,35] and seed quality [32,33]. Geren [33] investigated the effects of different N rates (0, 50, 75, 100, 125, 150, and 175 kg N ha^−1^) on quinoa seed yield and its components, which indicated a positive increase in seed yield by increasing N rates. This investigation indicated that for the Mediterranean conditions, a rate of 150 kg N ha^−1^ was appropriate to earn a 2.95 Mg ha^−1^ seed yield with a 16% protein concentration. However, Oelke et al. [36] suggested the possibility of a 4.5 Mg ha^−1^ seed yield in Colorado by increasing the N rate to 170 and 200 kg N ha^−1^ [36]. Likewise, Kaul et al. reported a 94% rise in quinoa seed yield by increasing the N rate to 120 kg N ha^−1^ [37].

Despite the influence of the increasing N rate on quinoa seed yield and its components in different soil types [38], the interaction effects between the N rates and other field practices such as irrigation could be important in different climate conditions [39,40]. Limited research on nitrogen fertilizer and irrigation water management is available in European humid weather conditions [38] or arid weather conditions in quinoa native regions [32]. For example, the results of Alandia et al. [35] showed that N might confer a certain degree of drought tolerance to quinoa as seed quality and yield of N-fertilized plants were not affected by drought stress. Their results under controlled conditions serve as a basis to elucidate drought tolerance mechanisms activated with N fertilization.

Soil organic matter content is lower in soil with high temperatures in arid and semi-arid weather conditions. Moreover, microbial activity is affected by soil water content [41]. Therefore, the nitrogen fertilizer application rate in arid climates is higher than those in humid climates. However, documented research on quinoa nitrogen fertilizer management in arid and semi-arid weather conditions in non-native regions (out of the Andes area) is not well documented. The complex cycle of nitrogen in the environment under different soil and weather conditions associated with nitrogen degradation and assimilation mechanism is an essential aspect of nitrogen fertilizer application [42]. Quinoa cultivation is becoming popular in non-native arid and semi-arid regions; however, there is still a gap in its yield response to different field management practices, especially in these climate conditions. Moreover, the rate of the optimum nitrogen fertilizer is still unknown under field conditions in non-native quinoa cultivation areas with water scarcity. Therefore, investigating the effect of different irrigation regimes and different nitrogen rates on quinoa yield, irrigation water use efficiency, and N use efficiency was the objective of this study.

## 2. Results

### 2.1. Water Use

Figure 1 illustrates the irrigation depth in three irrigation regimes along with reference evapotranspiration, mm, during the two growing seasons. Irrigation depths before applying irrigation regimes treatments were 326 mm and 256 mm with a 6-day interval in the first and second year, respectively. Irrigation treatments were initiated at the vegetative with bud formation stage. The highest value of irrigation depth belonged to full irrigation in both years (850 mm and 714 mm in the first and second years, respectively). Data showed that irrigation depths in 0.75 FI were 719 mm and 600 mm in both years, with a similar reduction of 15% and 15.7% compared to full irrigation. Likely, in 0.5 FI, irrigation depths were 588 mm and 485 mm, which decreased by 30% and 31.5% in the first and second years, respectively, compared to full irrigation (Figure 1).

There was no significant interaction between the effect of irrigation regimes and nitrogen application rate on the soil water contents before irrigation, according to the analysis of variance (*p* < 0.05). Furthermore, there was no significant difference in soil water contents between 0.75 FI and 0.5 FI, while they decreased by 8.2% compared to FI (Table 1). Furthermore, the mean soil water content in the control N treatment (0 kg N ha^−1^) was statistically higher than that in the 250 kg N ha^−1^. Nitrogen application at the rate of 250 kg N ha^−1^ noticeably decreased soil water contents before irrigation events by 4.5% compared to non-fertilized treatment due to higher crop growth and higher water uptake.

### 2.2. Seasonal Evapotranspiration

Actual crop evapotranspiration was estimated using water balance and averaged in two growing seasons, which is presented in Table 1. Overall, in the first year, reference potential evapotranspiration was higher than that in the second year due to higher temperature during the growing season (Figure 2); however, the difference was not statistically significant according to the analysis of variance (*p* < 0.05). The results indicated a significant difference amongst the irrigation strategies, in which FI had the highest ET (802.5 mm) followed by 0.5 FI with the lowest ET (528.0 mm) (Table 1). In non-fertilized treatment, ET decreased by 15% and 32% in 0.75 FI and 0.5 FI, respectively, compared to FI. Likewise, there was a reduction of 19% and 48% of ET in 0.75 FI and 0.5 FI in 250 kg N ha^−1^, respectively, compared to FI. Furthermore, nitrogen application did not show a significant impact on ET in both full and deficit irrigation regimes (Table 1).

### 2.3. Seasonal Evaporation and Transpiration

The mean value of soil surface seasonal evaporation over two growing seasons (Table 1) indicated that there was a significant difference in evaporation between the two nitrogen treatments (0 and 250 kg N ha^−1^) at all irrigation regimes. The nitrogen application rate of 250 kg N ha^−1^ remarkably decreased soil evaporation by 4.5%, on average, compared to non-fertilized treatment. Therefore, nitrogen application statistically reduced the soil evaporation during the growing season due to higher crop canopy development and higher coverage of the soil surface so that about 53–56.4% and 32–39% of ET was related to evaporation in non-fertilized and fertilized treatments, respectively. On the other hand, the evaporation did not significantly differ between FI and 0.75 FI (Table 1) in two nitrogen application rates, though both were different from the 0.5 FI. On average, deficit irrigation regimes decreased evaporation by 12% and 35% in 0.75 FI and 0.5 FI, respectively, compared to FI (Table 1). The two-year average of seasonal crop transpiration over the growing seasons is presented in Table 1. The highest value (479.3 mm) of crop transpiration was observed in the FI and 250 kg N ha^−1^, which was significantly higher than the other treatments. The application of 250 kg N ha^−1^ enhanced crop transpiration by 27, 34, and 54% compared to non-fertilized treatment in FI, 0.75, and 0.5 FI treatment, respectively (Table 1). Therefore, nitrogen application increased crop transpiration due to improving crop canopy growth, especially in treatments with water stress conditions. Additionally, deficit irrigation regimes as 0.75 FI and 0.5 FI remarkably decreased seasonal crop transpiration compared to FI, though they were not statistically different (Table 1). Generally, quinoa transpiration dropped by 19% and 30% in 0.75 FI and 0.5 FI, respectively, in comparison with that obtained in FI (Table 1).

### 2.4. Yield and Yield Components

There was a significant interaction between the effect of irrigation regimes and nitrogen application rate on quinoa seed yield, total dry matter, and harvest index (*p* < 0.05). Therefore, the interactive effects are presented in Table 2.

#### 2.4.1. Seed Yield

Quinoa seed yield was significantly different between the two years (2017–2018) according to the analysis of variance (*p* < 0.05); therefore, the measured data were separately presented and discussed (Table 2). In the first year, seed yield dramatically decreased by 80% compared to that obtained in the second year. The highest seed yield was 0.96 Mg ha^−1^ in the first year, which was 80% lower than that obtained in the second year. This dramatic decrease in the first year occurred due to facing air temperatures higher than 35 °C in the flowering stage (Figure 2).

In the first year, the effects of different irrigation regimes and nitrogen rates on quinoa seed yield were significant (Table 2). Nitrogen application increased seed yield in FI, and the highest value was obtained in 375 kg N ha^−1^ while there was no significant difference in seed yield between 375 kg N ha^−1^ and 250 kg N ha^−1^ in the first year with pest damage and unfavorable air temperature. However, in the second year with favorable air temperature, the highest yield with a significant increase was obtained in 375 kg N ha^−1^ and FI regime. In the deficit irrigations, on the other hand, the highest value of seed yield was achieved at 250 kg N ha^−1^, and adding more N declined the quinoa seed yield (Table 2). Deficit irrigation as 0.75 FI and 0.5 FI reduced seed yield by 15% and 30%, respectively, compared to FI. In comparison with FI, applying the 0.5 FI regime decreased quinoa seed yield by 38.9%, 35.7%, 17.9%, and 33% in the nitrogen application rates at 0, 125, 250, and 375 kg N ha^−1^, respectively. Even though seed yield was much higher in the second year than that in the first year, the same trend in the results was obtained in the second year, which indicated the significant effect of different N application rates and irrigation strategies on quinoa seed yield (Table 2). Despite the first year, the interaction effect of N application rate and irrigation regimes was significant in the second year (*p* < 0.05). In FI, the highest seed yield was obtained in 375 kg N ha^−1^ (4.71 Mg ha^−1^), followed by 250 kg N ha^−1^ (4.27 Mg ha^−1^), whereas the maximum seed yield was obtained in 250 kg N ha^−1^ in 0.75 FI and 0.5 FI as 4.12 Mg ha^−1^ and 3.58 Mg ha^−1^, respectively. Although various irrigation regimes showed a significant difference in seed yield, N application rates of 250 and 375 kg N ha^−1^ did show any significant difference in seed yield. The application of 125 kg N ha^−1^ enhanced quinoa seed yield by 42.5% compared to non-fertilized treatment, whereas applying 375 kg N ha^−1^ increased seed yield by 69%. A reduction of 53% and 16% in seed yield was observed in 0.5 FI with 0 kg N ha^−1^ and 250 kg N ha^−1^ application rates compared to 375 kg N ha^−1^, respectively. Likely, 0.5 FI with 125 kg N ha^−1^ and 375 kg N ha^−1^ decreased seed yield by 32.7% compared to FI (Table 2). Generally, increasing N fertilizer to 250 kg N ha^−1^ noticeably enhanced the quinoa seed yield; therefore, statistically, it could be the proper rate of N application in rising seed yield for deficit irrigation regimes. However, under non–limited water resource conditions, an FI and N application rate of 375 kg ha^−1^ could be used for higher seed yield.

#### 2.4.2. Total Dry Matter

Quinoa total dry matter (TDM) was not significantly different in the two years (*p* < 0.05). Therefore, the mean values of TDM over two years are presented in Table 2. The interaction effect between N application rates and irrigation regimes was significant (*p* < 0.05). The highest value of TDM was obtained in FI at the rate of 375 kg N ha^−1^ (9.03 Mg ha^−1^), while in deficit irrigation regimes, 250 kg N ha^−1^ had the highest TDM values (Table 2). However, there was no significant difference between 250 kg N ha^−1^ and 375 kg N ha^−1^ in TDM in FI for all irrigation regimes. When the N application rate exceeded 250 kg N ha^−1^, it caused a rise of 40.5% in TDM compared to non-fertilized treatment. In deficit irrigations (0.75 FI and 0.5 FI), TDM matter did not show a significant difference when the N rate exceeded 125 kg N ha^−1^. However, considering the seed yield increase, the N rate higher than 250 kg N ha^−1^ did not show a significant effect on seed production. Furthermore, a linear relationship between quinoa TDM and total transpiration during the growing season, indicated that increasing transpiration directly increased total dry matter (TDM = 0.0188 T, R^2^ = 0.92, *p* < 0.001, SE = 0.45, n = 18). This relationship can be used for comparing quinoa transpiration efficiency with other cereals.

#### 2.4.3. Harvest Index

A reduction of 80% in quinoa seed yield occurred in the first year as compared with the second year. This was due to high air temperature in the flowering stage and pests damage in the first year compared to the second year. Therefore, the harvest index was determined only for the second year (Table 2). The results indicated that the interaction effect between N application rate and irrigation regimes was significant, as well as the main effects (*p* < 0.05). The harvest index varied between 0.31 and 0.45 for quinoa in the second year. Although 0.75 FI in all fertilized treatments did not show a significant difference in harvest index with those in FI, harvest index significantly decreased in 0.5 FI and N rates by 25.5%, 17.8%, and 11% in non-fertilized, 125 kg N ha^−1^, and 375 kg N ha^−1^, respectively, compared to FI. However, the harvest index in 0.5 FI and 250 kg N ha^−1^ were statistically the same as that in FI (Table 2). Overall, the N application effectively increased harvest index in comparison with that obtained in non-fertilized treatment, especially in 0.5 FI. In 0.75 FI, N application of 250 kg N ha^−1^ and 375 kg N ha^−1^ raised harvest index by 12.5% and 10%, respectively, compared to non-fertilized treatment. Likewise, 0.5 FI increased the harvest index by 19.4%, 38.7, and 29% in N application rates of 125, 250, and 375, respectively. Taking the second-year harvest index into account, quinoa seed yield for the first year was estimated by multiplying HI and first-year dry matter (Table 2). Therefore, seed yield was estimated by omitting pest and heat damage to seed yield in the first year. According to this result, the highest seed yield (3.44 Mg ha^−1^) was obtained in FI with 375 kg N ha^−1^ and the lowest value (1.22 Mg ha^−1^) in 0.5 FI with non-fertilized treatment (Table 2).

### 2.5. Total Nitrogen Uptake

The total N uptake is estimated by seed N uptake plus straw N uptake (Table 2). The analysis of variance showed that the interaction effects between irrigation regimes and N application rates on total N uptake were significant (*p* < 0.05). Generally, a high-water deficit (0.5 FI) resulted in a high reduction in total N uptake (Table 2). In FI, increasing the N rate from 250 to 375 kg N ha^−1^ significantly raised total N uptake by 17%, whereas this increase was not significant in 0.75 FI and 0.5 FI. In non-fertilized treatment, irrigation application of 0.75 FI and 0.5 FI dropped total N uptake by 13.5% and 48% compared to FI. Likewise, in 375 kg N ha^−1^, the application of 0.75 FI and 0.5 FI decreased total N uptake compared to FI by 20.3% and 30.5%, respectively. Furthermore, increasing N application rates significantly increased total N uptake (Table 2). For example, the application of 125 kg N ha^−1^ compared to non-fertilized treatment significantly enhanced total N uptake by 34%, 41%, and 95.9% in FI, 0.75 FI, and 0.5 FI, respectively. Therefore, the highest value of total N uptake was obtained in 375 kg N ha^−1^ and FI, while the lowest value was obtained in non-fertilized treatment and 0.5 FI.

### 2.6. Residual Soil NO_3_-N

The residual soil NO_3_-N in three different depths, including, 0–30 cm, 30–60 cm, and 60–90 cm, was measured before sowing and after harvest in both years, and the total residual NO_3_-N, in the soil profile is presented in Table 2. The results indicated that the interaction effect of irrigation regimes and N application rates on soil NO_3_-N was statistically significant (*p* < 0.05). In non-fertilized conditions, an irrigation regime of 0.75 FI noticeably enhanced the residual soil NO_3_-N by 62% compared to FI, while, in fertilized treatments, there was no significant difference between 0.75 FI and FI in residual soil NO_3_-N (Table 2). By contrast, 0.5 FI significantly increased residual soil NO_3_-N in all N rates compared to FI and 0.75 FI. In this study, the highest residual soil NO_3_-N was obtained in 0.5 FI and 375 kg N ha^−1^ when the FI and non-fertilized treatment had the lowest value (Table 2). Furthermore, measured residual soil NO_3_-N at different soil depths after harvest averaged in both years is presented in Figure 2. This figure illustrates the variation of soil NO_3_-N in all treatments after harvest compared to that before planting. In FI, residual soil NO_3_-N in all N application rates at the topsoil layer (0–30 cm) decreased at the end of the growing season (Figure 3). Although, it increased at deep layers (60–90 cm), especially at higher N rates (250 kg N ha^−1^ and 375 kg N ha^−1^) due to N leaching to the lower soil layers. Application of 0.75 FI led to NO_3_-N accumulation at upper soil layers compared to that in FI. A dramatic increase was observed at the 30–60 cm soil layer by applying 375 kg N ha^−1^. When the irrigation water decreased to a lower level as 0.5 FI, NO_3_-N accumulation in the middle of the soil profile (30–60 cm) increased, which remarkably exceeded the initial soil NO_3_-N at topsoil (0–30 cm) and 60–90 cm in two application N rates of 250 kg N ha^−1^ and 375 kg N ha^−1^. Therefore, a higher reduction in irrigation water led to a much higher increase in residual soil NO_3_-N, almost the same as increasing N application rate effects on residual soil NO_3_-N.

### 2.7. The Relationship between Total N Uptake and Transpiration

Due to the influential role of seasonal transpiration (T) in N uptake (TNU), the relationship between these parameters was obtained for two application N rates of 0 kg N ha^−1^ and 250 kg N ha^−1^ as follows:
TNU = 0.195 T for 0 N ha^−1^ nitrogen rateR^2^ = 0.76, *p* < 0.001, SE = 0.35, n = 9(1)
TNU = 0.16 T + 41.4 for 250 N ha^−1^ nitrogen rateR^2^ = 0.75, *p* < 0.001, SE = 0.40, n = 9(2)
where TNU and T are total nitrogen uptake (kg ha^−1^) and seasonal transpiration (mm), respectively. It is indicated that passive absorption of N in the plant is regulated by transpiration. Therefore, transpiration is an effective force in N uptake in two N application rates of 0 kg N ha^−1^ and 250 kg N ha^−1^. However, the slope of T in the regression is higher in the N application rate of 0 kg N ha^−1^. According to Equation (2), at an N application rate of 250 kg ha^−1^ about 41.4 kg N ha^−1^ [intercept of Equation (2)] is absorbed by active absorption from the root zone. It is indicated that at high N application rates, both N absorption mechanism is effective [Equation (2)].

### 2.8. The Relationship between Soil Total Available N and Yield

Due to the influential role of soil total available N in producing crop yield and N uptake, the relationships between these parameters were obtained (Figure 3). Soil total available N (STAN) included soil N, applied N by fertilizer, and N applied by irrigation water. Soil N was considered to be residual N at planting plus the N mineralization that was determined considering 3% of the soil organic N mineralized to inorganic N [43].The relationship between seed yield and soil total available N indicated that in all irrigation regimes, seed yield increased gradually by increasing the soil total available N and reached a maximum value at a specific value of soil total available N, after which it dropped (Figure 4a). A similar trend was found for the total dry matter and crop N uptake (Figure 4b,c). The maximum value of the soil total available N was different in various irrigation regimes. Therefore, the maximum values to reach the highest seed yield were obtained by numerical differentiation of the equation for each irrigation regime. These values are 822, 639, and 634 kg N ha^−1^ in FI, 0.75 FI, and 0.5 FI, respectively. By subtracting applied N by irrigation water, mineralized N, and soil N before planting, the optimum amount of N fertilizer was individually estimated for different irrigation strategies. In FI, 0.75 FI, and 0.5 FI, the maximum N fertilizer rate in the study area is estimated as 467, 287, and 285 kg N ha^−1^, respectively. Overall, seed yield, total dry matter, and N uptake decreased when applied water dropped from full irrigation to 0.5 FI. Considering the total available soil N and irrigation water depth, a relationship was found to predict quinoa seed yield as follows:SY = −1.758 × 10^−5^ N^2^ − 1.143 × 10^−5^ I^2^ + 23.22 × 10^−3^ N + 0.0192 I − 10.96R^2^ = 0.88, *p* < 0.001, SE = 0.22, n = 9(3)
where SY, N, and I are the seed yield (Mg ha^−1^), soil total available N (kg ha^−1^), and irrigation water depth (mm), respectively. This relationship could be used in modeling to predict seed yield based on soil total available N and applied water. Further, contour (iso-quant) plots were developed in (Figure 5) to show the combined effect of soil total available N and irrigation water depth on seed yield for practical use by farm managers. The quadratic equation implied that the optimum N application rate would be different in various irrigation regimes to reach the maximum seed yield.

### 2.9. Water Use Efficiency

The analysis of variance showed a significant effect of the interaction between irrigation regimes and the N application rates on the seed water use efficiency (SWUE) (*p* < 0.05). Table 1 presents the comparison between means of SWUE for two N application rates at different irrigation regimes. Crop evapotranspiration was calculated from Equation (4) to determine WUE. As the soil water content was measured by neutron probe tubes located at two N application rates (0 and 250 Kg N ha^−1^), WUE is presented for these two treatments. Generally, the N application significantly raised SWUE and DWUE. Adding 250 kg N ha^−1^ noticeably increased SWUE by 42%, 67%, and 165.5% in FI, 0.75 FI, and 0.5 FI, respectively. There was a significant difference between FI and deficit irrigation regimes in SWUE when the 250 kg N ha^−1^ rate was applied, whereas no significant difference was observed between 0.75 FI and 0.5 FI. By contrast, non-fertilized treatment and 0.5 FI significantly decreased SWUE by 31% compared to 0.75 FI (Table 1). Furthermore, total dry matter water use efficiency (DWUE) was determined at different irrigation regimes and two N application rates (Table 1). The highest DWUE of 1.8 kg m^−3^ was obtained in 0.5 FI with the application of 250 kg N ha^−1^, whereas it was 19% higher than those obtained in FI and 0.75 FI. In non-fertilized treatment, DWUE decreased by 12% in 0.5 FI compared to those obtained in FI and 0.75 FI. Accordingly, increasing the N application rate to 250 kg N ha^−1^ enhanced DWUE by 42%, 43%, and 94% in FI, 0.75 FI, and 0.5 FI, respectively. Generally, deficit irrigation could not increase SWUE and DWUE in non-fertilized treatment.

### 2.10. Nitrogen Efficiencies

The results of the estimated nitrogen use efficiency (NUE) and nitrogen yield efficiency (NYE) at different irrigation regimes and N application rates in the second year are presented in Table 3. There is no significant interaction between the effect of irrigation regimes and nitrogen application rate on nitrogen use efficiencies (*p* < 0.05). However, the main effects of the irrigation regimes and nitrogen rates had a significant effect on both NUE and NYE (*p* < 0.05). In general, the increasing N application rate decreased NUE though there was no significant difference between 125 kg N ha^−1^ and 250 kg N ha^−1^. Increasing the N application rate from 125 kg N ha^−1^ and 250 kg N ha^−1^ to 375 kg N ha^−1^ noticeably decreased NUE by 29% and 21%, respectively (Table 3). Reducing water application to half (0.50 FI) significantly increased NUE, whereas 0.5 FI increased NUE by 19% and 29% compared to FI and 0.75 FI, respectively. Same as the NUE, the increasing N rate decreased NYE with no significant difference between 125 kg N ha^−1^ and 250 kg N ha^−1^. The nitrogen application rate of 375 kg N ha^−1^ dropped NYE by 44.3% and 34% compared to 125 kg N ha^−1^ and 250 kg N ha^−1^, respectively (Table 3). In addition, no significant difference was found between 0.75 FI and FI in NYE, while the application of 0.5 FI increased NYE by 54% compared to FI. Table 3 presented the mean physiological N efficiency (NPE) at different irrigation regimes and N application rates. Analysis of variance showed that the effect of irrigation regimes was the only significant factor in NPE (*p* < 0.05). Compared to FI, deficit irrigation of 0.75 FI significantly increased NPE by 26.7%, whereas 0.5 FI raised NPE by 17%, which was not statistically different from FI.

## 3. Discussion

### 3.1. Quinoa Yield Is Affected by N Application Rate and Irrigation Regimes

Comparing the results of quinoa seed yield in two growing seasons, apart from the irrigation and N application rates, showed an unexpected drop in seed yield (0.96 Mg ha^−1^) in the first year. However, TDM was not different in the two growing seasons. Therefore, the results from these experiments could be generalized for field management decisions in arid and semi-arid weather conditions. The dramatic decrease in the first year occurred due to the late planting dates in the first year and air temperatures higher than 35 °C in the flowering stage (Figure 2). The value of 35 °C for air temperature is considered the threshold value for quinoa in Algeria, Lebanon, Yemen, and Iraq during the flowering stage [6,44,45,46,47]. This could be due to the strong dependence of pollen moisture on-air vapor pressure deficit [48]. Therefore, a slight temperature increase higher than 35 °C would greatly reduce seed yield [49]. The increasing N application rate remarkably raised seed yield, and the highest seed yield in the second year was 4.7 Mg ha^−1^ in 375 kg N ha^−1^ with FI (Table 2), which is inconsistent with findings by Shams [50], and Kakabouki et al. [2] who reported the rise of quinoa seed yield with increasing N application rate to 120, 200, and 360 kg N ha^−1^. Additionally, in the weather conditions of Turkey, the highest seed yield, ranged between 4.1 Mg ha^−1^ and 8.7 Mg ha^−1^ [39], whereas it was 3.3 Mg ha^−1^ in the south of Morocco [51]. Deficit irrigation (0.5 FI) significantly reduced quinoa seed yield, while there was no significant difference between FI and 0.75 FI. Therefore, an irrigation regime of 0.75 FI is suggested in the case of scarce water. Similar findings were reported by Yazar et al. [52]. According to our findings, in water scarcity conditions, 0.75 an FI and N application rate of 250 kg N ha^−1^ is recommended.

Considering the total dry matter, increasing the N application rate higher than 125 kg N ha^−1^ did not show a significant increase in the dry matter; however, the highest value of DM (9.03 Mg ha^−1^) was obtained in 375 kg N ha^−1^ (Table 2). Similarly, Kakabouki et al. [2] reported no significant difference in the dry matter between two N application rates of 200 kg N ha^−1^ and 345 kg N ha^−1^. In the Mediterranean conditions, the highest average DM was reported as 8.5 kg ha^−1^ [51]. The linear relationship between quinoa TDM and total transpiration during the growing season indicated that increasing transpiration directly increased total dry matter (TDM = 0.0188 T). The comparison between the slope of the linear relationship between quinoa total dry matter and seasonal transpiration with that reported by Azizian and Sepaskhah [53] for maize as a C_4_ crop (0.028 Mg ha^−1^) and Bahari-Sadi for saffron as a C_3_ crop (0.010 Mg ha^−1^) indicated that maize could produce higher TDM in given seasonal transpiration as compared with quinoa as a C_3_ crop. However, quinoa can produce higher TDM in given seasonal transpiration as compared with saffron in similar weather conditions. In general, based on our findings, it is recommended to use 375 kg N ha^−1^ for reaching the optimum quinoa yield in full irrigation (no water scarcity) with an average of 24% soil water content before irrigation, and 250 kg N ha^−1^ is required for 0.75 FI and 0.5 FI (scare water conditions) with an average of 21.0 and 20.8% soil water content before irrigation, respectively.

### 3.2. Water and N Efficiencies as Influenced by N Application Rate

In the current study, the highest water use efficiency obtained in 0.5 FI with 250 kg N ha^−1^ was 0.77 kg m^−3^, which was lower than 1.2 kg m^−3^ obtained in Italy [54]. Increasing N rates resulted in higher water use efficiency; however, the inverse impact of exceeding the N application rate from 250 kg N ha^−1^ was observed (Table 1). Similarly, increasing the N application rate reduced all the N efficiencies (NUE, NYE, and NPE), which is in agreement with the findings of Kakabouki et al. [2] for quinoa and Mehrabi and Sepaskhah [55] for winter wheat. Therefore, N application at a rate of 250 kg N ha^−1^ could be suggested as the optimum rate, reaching the highest water use efficiency and N use efficiencies in the study region. Although an increase was found in the water and N use efficiencies in the application of 0.5 FI with 250 kg N ha^−1^, it is not recommended to produce quinoa crops because it might not be economic, as it remarkably decreases seed yield.

### 3.3. Nitrogen Uptake Mechanism and N Application Rate

Nutrient ion’s movement from the soil solution to the vascular center of the root cell membrane may be passive or active. Passive absorption is the movement across a membrane from higher to lower concentrations. A similar gradient leads to crop transpiration. Active absorption requires metabolical energy. As with the uptake of other nutrients, N uptake activities are both strongly regulated by a high plant N status [56]. A different nitrogen uptake mechanism is discussed by [57,58,59] in the physiological and molecular mechanism. According to our findings, passive absorption is an effective force in N uptake in two N application rates of 0 kg N ha^−1^ and 250 kg N ha^−1^. However, at high N application rates, both passive and active N absorption mechanism is effective.

## 4. Materials and Methods

### 4.1. Experimental Site and Design

The experimental research was conducted in the Experimental Research Station of the Agricultural College, Shiraz University, Shiraz, Iran, during two growing seasons (2017 and 2018). The station is located at 29°56′ N, 52°02′ E and is 1810 m above sea level, with a semi-arid climate. Maximum and minimum temperatures (T_max_, T_min_), average relative humidity, and rainfall during both growing seasons are presented in Figure 6. T_max_ and T_min_ during the first growing season were, on average, 32.4 °C and 12 °C, respectively. In contrast, the mean daily T_max_ and T_min_ during the second growing season were 27 °C and 7 °C, respectively. The rainfall depths during the two growing seasons were 29 and 64 mm, respectively.

The experimental design was a factorial arrangement with randomized complete blocks in three replications. The treatments consisted of four nitrogen application rates of 0, 125, 250, and 375 kg N ha^−1^ (N1, N2, N3, and N4, respectively) and three levels of irrigation water regimes. The irrigation regimes included full irrigation (FI), 75% of full irrigation (0.75 FI), and 50% of full irrigation (0.5 FI). Therefore, 36 plots were constructed with the dimensions of 2 m × 2 m and placed at a distance of 1.5 m from each other (Figure 6a). The physical and chemical properties of soil adapted from Yarami and Sepaskhah (2015) are presented in Table 4 [60].

After a deep plowing (30 cm) during the land preparation, triple superphosphate (CaH_2_PO_4_, including 46% P_2_O_5_) at a rate of 30 kg ha^−1^ was mixed with the soil surface layer. The quinoa seeds (Titicaca cultivar, developed by University of Copenhagen, Denmark) were planted in 6 rows with 0.33 m spacing. The seeds were placed in 0.01–0.02 m depth with a 0.15 m distance on the row. The sowing dates were 6 April and 3 March in the first and second growing seasons, respectively, with the average air temperature of 10 to 12 °C. Two weeks after germination, the crops were hand-thinned to obtain a uniform density of 20 plants per plot. Physical weed control was also conducted every two weeks during the growing seasons. Nitrogen as urea (46% N) was added to the soil surface at the vegetative and seed filling stages (Figure 6b), which were 60, and 80 days after sowing, respectively.

### 4.2. Irrigation Requirement

The irrigation water depth was determined using the daily potential evapotranspiration and crop coefficient for quinoa. The potential evapotranspiration was estimated using the modified FAO-Penman–Monteith method [61]. The crop coefficients at initial, mid-season, and late season growth stages for quinoa were also determined by Talebnejad and Sepaskhah as 0.58, 1.2, and 0.8, respectively [14]. Therefore, irrigation depth as crop evapotranspiration (ET_c_) was determined by multiplying crop coefficient and potential evapotranspiration during the irrigation intervals. The irrigation treatments were initiated at the vegetative with bud formation stage with a 7-day interval as surface basin irrigation. The total irrigation water depths were 850 and 714 mm for the two growing seasons, respectively.

### 4.3. Crop Actual Evapotranspiration

Crop actual evapotranspiration (ET) during each growing season is estimated by following the soil water balance equation considering negligible deep percolation:(4)ET=I+P±ΔS
where I and P are the irrigation and precipitation depth (cm), respectively. ΔS implies soil water variations (cm) in each interval.

The soil water content was measured by a neutron probe three times during the growing season at four soil depths (0–30 cm, 30–60 cm, 60–90 cm, and 90–120 cm) for two nitration application rates (0 and 250 kg N ha^−1^). Soil evaporation was measured using micro-lysimeters [55], which were installed in the plots with different irrigation regimes and two N treatments (N1 and N3) with three replications. Micro-lysimeters were made of a small cylinder with 10 cm diameter and 30 cm height filled up with the same field soil (Figure 6c). The micro-lysimeters were weighted between irrigation intervals. Then, decreasing in micro-lysimeter weight was divided per the micro-lysimeter area to determine the evaporation from the soil surface.

### 4.4. Field Measurements

Characterizing the plants in the fields needs to be managed with attention considering the last international consensus on quinoa phenotyping methodologies [62]. Ten to fourteen days after the last irrigation event, four rows in the middle of each plot were harvested [134 and 122 days after planting (17 August and 3 July) in the first and second year, respectively]. The panicles were separated, and the seeds and shoots were oven-dried at 72 °C for about 48–72 h to determine seed yield, shoot dry weight, and total dry matter. The harvest index was calculated as seed yield divided by the total top dry matter. The seed and shoot nitrogen concentrations were measured by the Kjeldahl method then nitrogen uptake was determined by multiplying seed and shoot N concentrations by their relevant dry weight. Before N application and after harvesting, soil samples were taken from three depths, 0–30 cm, 30–60 cm, and 60–90 cm, and air-dried to determine the residual soil nitrate spectrophotometrically.

### 4.5. Water and Nitrogen Use Efficiencies

Seed water use efficiency (SWUE) and dry matter water use efficiency (DWUE) are determined as the ratio between seed yield (SY) or total dry matter (TDM) and crop water use (ET). Furthermore, nitrogen use efficiency as a vital component in obtaining net yield with high quality has different descriptions such as nitrogen use efficiency (NUE), nitrogen yield efficiency (NYE), and physiological N efficiency (NPE) [63] as follows:(5)$$\mathrm{NUE}=\frac{NU_{a}-NU_{c}}{NR_{a}-NR_{c}}$$
(6)$$\mathrm{NYE}=\frac{Y_{a}-Y_{c}}{NR_{a}-NR_{c}}$$
(7)$$\mathrm{NPE}=\frac{Y_{a}-Y_{c}}{NU_{a}-NU_{c}}$$
where *NU_a_, Y_a_*, and *NR_a_* are the quinoa N uptake, quinoa seed yield, and nitrogen application rate, kg ha^−1^, respectively, in different treatments. While *NU_c_, Y_c_*, and *NR_c_* are the quinoa N uptake, seed yield, and nitrogen application rate, kg ha^−1^, respectively, in the control treatment (N application rate of 0 kg ha^−1^).

### 4.6. Statistical Analysis

Statistical analysis was carried out using software SAS 9.4 software (SAS Institute Inc., Carry, NC, USA). The interaction effects between irrigation regimes and N application rates were evaluated using the analysis of variance (ANOVA), and the year was incorporated into the model. The means were compared at a 5% level of probability using Duncan’s multiple range tests.

## 5. Conclusions

The nitrogen fertilizer application rate in arid and semi-arid areas is higher due to warmer climates. However, deficit irrigation improves NUE. Therefore, the effect of different irrigation regimes and nitrogen rates on quinoa yield is a challenging issue in those areas. Increasing nitrogen fertilizer application levels from 250 to 375 kg N ha^−1^ under FI and deficit irrigation (0.75 FI and 0.5 FI) did not cause a significant difference in grain yield and the total dry matter of quinoa. Furthermore, the application of 0.75 FI led to NO_3_-N accumulation in upper soil layers compared to that in FI, which facilitated nitrogen uptake and reduced nitrate loss to deeper layers of the soil. Therefore, an appropriate nitrogen application rate of 250 kg N ha^−1^ and deficit irrigation of 0.75 FI is suggested as the optimum management in the study area, with challenging water scarcity based on SWUE, NUE, and total nitrogen uptake. However, in the area with ample water resources, FI with 375 kg, N ha^−1^ could be recommended based on seed yield and total nitrogen uptake. These factors should be considered as struggling with the potential quinoa seed yield and considering fertilizer application environmental impacts. Questions about which environmental factors impressively restricted the quinoa growth for optimizing the potential yield need further investigations.

## Figures and Tables

**Figure 1 plants-11-02048-f001:**
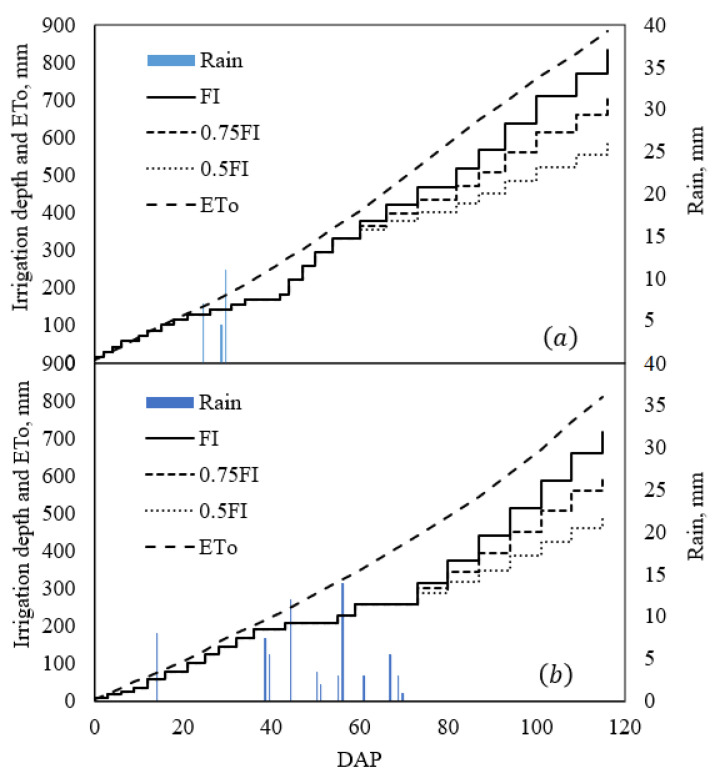
Irrigation depth, ET_o_, and rainfall during the quinoa growing season; (**a**) first year; (**b**) second year. DAP: days after planting.

**Figure 2 plants-11-02048-f002:**
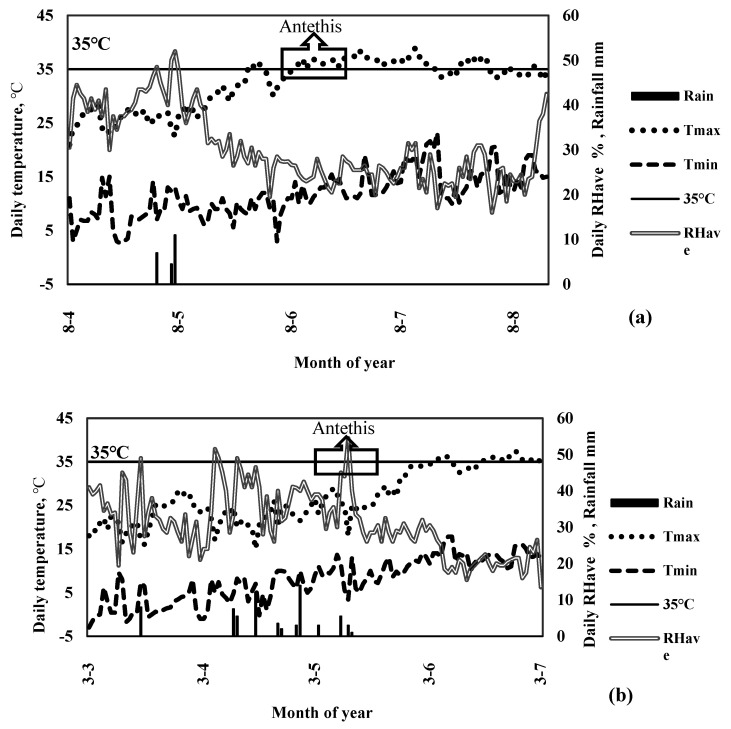
Daily maximum and minimum air temperature (T_max_, T_min_), relative humidity (RHavg), and rainfall during both growing periods. (**a**) 2017; (**b**) 2018.

**Figure 3 plants-11-02048-f003:**
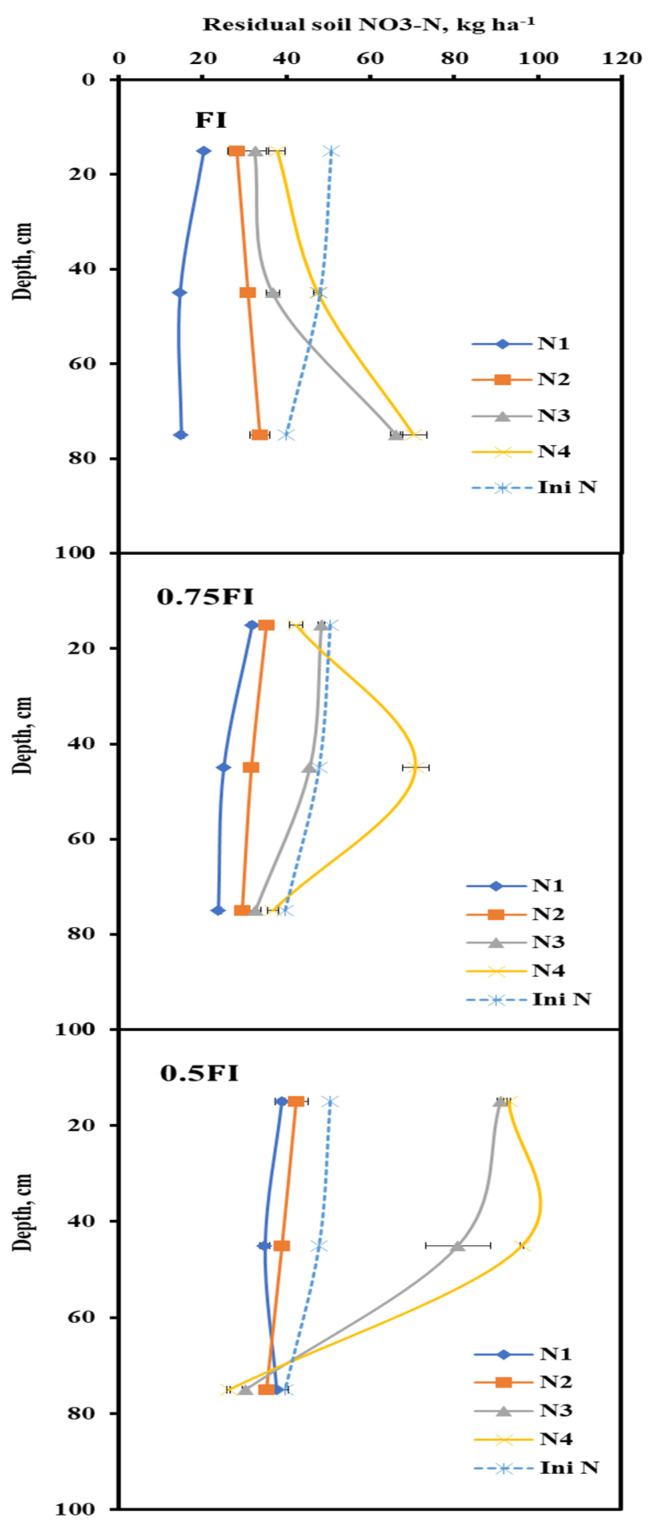
Mean residual soil NO_3_-N at different soil depths after harvest over two years.

**Figure 4 plants-11-02048-f004:**
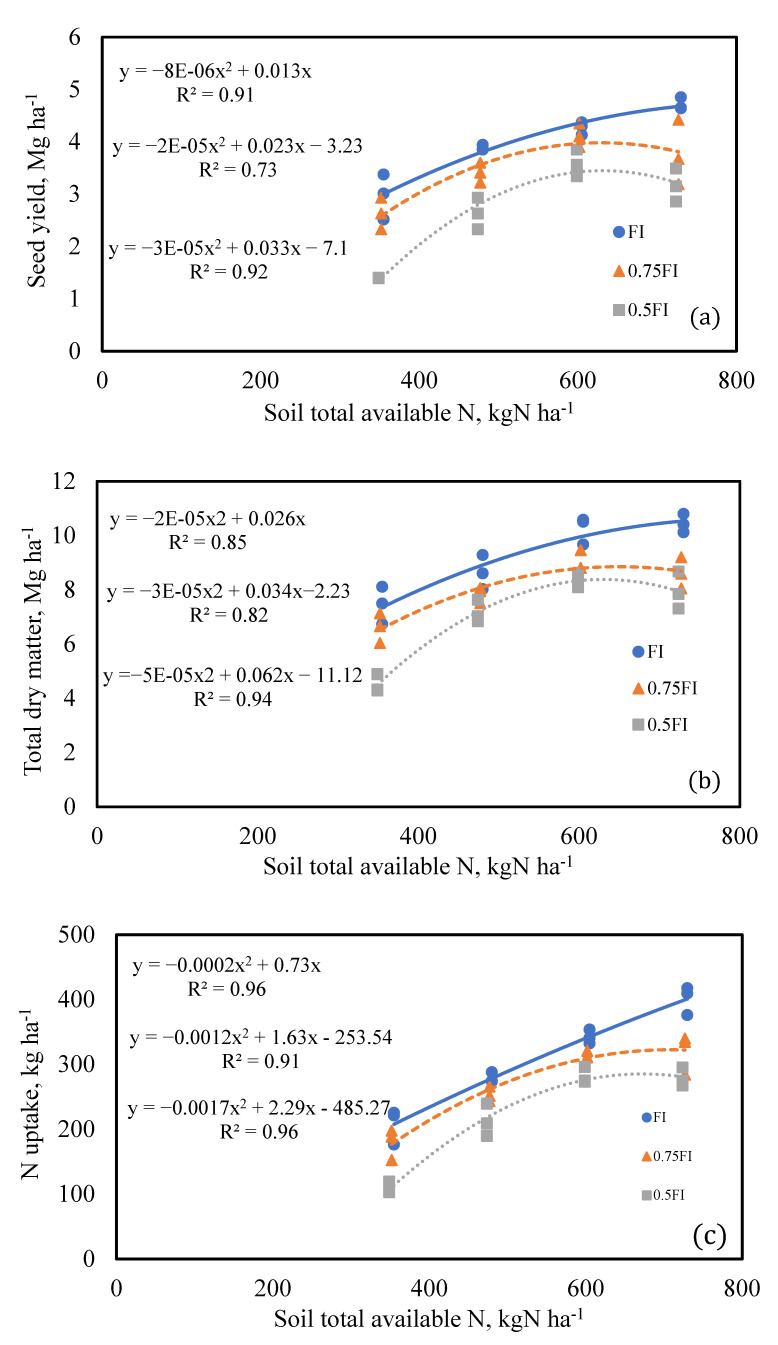
The relationship between soil total available N and quinoa seed yield, dry matter, and crop N uptake.

**Figure 5 plants-11-02048-f005:**
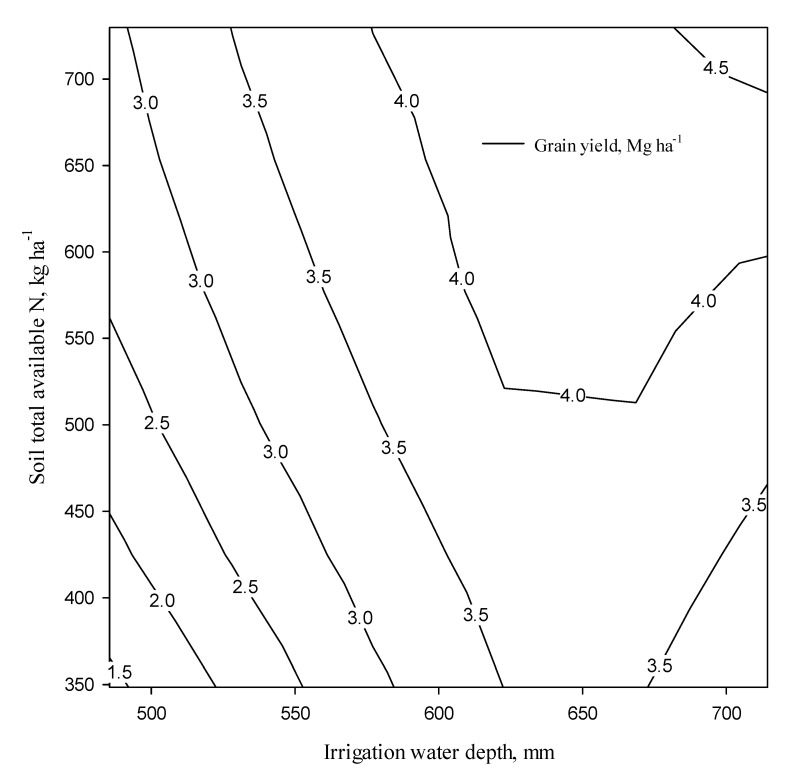
Nomo graph showing the relationship between irrigation water depth and soil total available N for obtaining different quinoa seed yields.

**Figure 6 plants-11-02048-f006:**
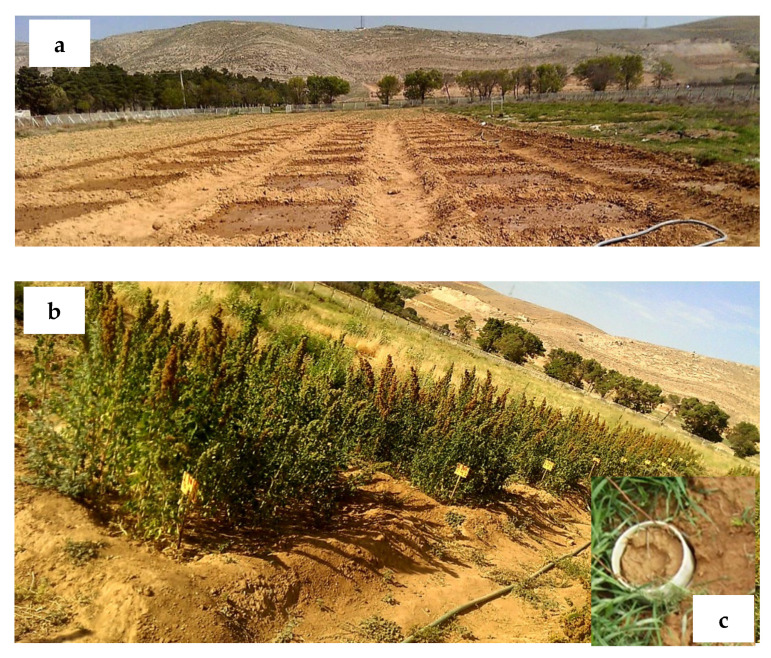
Experimental plot design at planting date (**a**), quinoa at seed filling stage (**b**) and the micro-lysimeter (**c**) for soil water evaporation measurement.

**Table 1 plants-11-02048-t001:** Seasonal mean soil water content, seasonal evapotranspiration, evaporation, transpiration, seed, and total dry matter water use efficiencies (SWUE, DWUE) for two nitrogen application rates and irrigation regimes averaged in two growing seasons.

Parameters	Irrigation Treatment	Nitrogen Application Rate (kg N ha^−1^)	
		0	250
Mean volumetric soil water content before irrigation, %	FI	23.8 a *	22.5 ab
	0.75 FI	22.2 ab	21.0 b
	0.5 FI	21.2 b	20.8 b
Seasonal evapotranspiration, mm	FI	802.50 a	782.1 a
	0.75 FI	676.5 b	655.9 b
	0.5 FI	542.2 c	528.0 c
Seasonal transpiration, mm	FI	377.4 b	479.3 a
	0.75 FI	295.7 c	396.2 b
	0.5 FI	236.2 c	363.8 b
Seasonal evaporation, mm	FI	425.1 a	302.8 b
	0.75 FI	380.9 a	259.7 b
	0.5 FI	306.0 b	164.2 c
SWUE, kg m^−3^	FI	0.43 c	0.61 b
	0.75 FI	0.42 c	0.70 a
	0.5 FI	0.29 d	0.77 a
DWUE, kg m^−3^	FI	1.04 c	1.48 b
	0.75 FI	1.08 c	1.54 b
	0.5 FI	0.93 d	1.80 a

* Means followed by the same letter in each trait are not significantly different at a 5% level of probability.

**Table 2 plants-11-02048-t002:** Seed yield (Mg ha^−1^), the two-year mean of total dry matter (Mg ha^−1^), harvest index and estimated seed yield in the first year’, hypothetically discarding the pest and unfavorable air temperature (Mg ha^−1^), total N uptake, (kg N ha^−1^), and residual soil NO_3_-N, (kg N ha^−1^) in different N application rates and irrigation regimes.

Parameters	Irrigation Treatment	Nitrogen Application Rate (kg N ha^−1^)			
		0	125	250	375
Seed yield, Mg ha^−1^ (2017)	FI	0.77 cde *	0.84 bc	0.89 ab	0.96 a
	0.75 FI	0.55 gh	0.68 ef	0.85 bc	0.8 bcd
	0.5 FI	0.47 h	0.54 h	0.73 def	0.64 fg
Seed yield, Mg ha^−1^ (2018)	FI	2.96 gf	3.91 bcd	4.27 b	4.71 a
	0.75 FI	2.63 g	3.41 def	4.12 bc	3.76 cd
	0.5 FI	1.39 h	2.63 g	3.58 de	3.17 ef
Total dry matter, Mg ha^−1^	FI	6.37 bc	7.43 abc	8.86 a	9.03 a
	0.75 FI	5.79 c	6.81 bc	7.87 ab	7.40 abc
	0.5 FI	4.23 d	6.31 bc	7.19 bc	6.91 bc
Harvest index	FI	0.40 bc	0.45 a	0.42 ab	0.45 a
	0.75 FI	0.40 bc	0.44 ab	0.45 a	0.44 ab
	0.5 FI	0.31 d	0.37 c	0.43 ab	0.40 bc
Estimated seed yield in the first year (2017), Mg ha^−1^	FI	2.10 gh	2.83 cd	3.13 b	3.44 a
	0.75 FI	1.98 h	2.51 ef	3.08 bc	2.68 de
	0.5 FI	1.22 i	2.01 h	2.59 def	2.35 fg
Total N uptake, kg N ha^−1^	FI	96.56 de	129.61 c	159.15 b	186.49 a
	0.75 FI	83.53 e	117.85 c	147.68 b	148.57 b
	0.5 FI	50.49 f	98.90 d	130.82 c	129.71 c
Residual soil NO_3_-N, kg N ha^−1^	FI	49.98 h	92.59 f	135.61 d	155.59 c
	0.75 FI	81.01 g	96.81 f	127.05 d	150.56 c
	0.5 FI	111.94 e	117.22 e	202.96 b	216.26 a

* Means followed by the same letter in each trait are not significantly different at the 5% level of probability.

**Table 3 plants-11-02048-t003:** Nitrogen use efficiency (NUE) and nitrogen yield efficiency (NYE) and physiological N efficiency (NPE) at different irrigation regimes and N application rates in the second year.

	NUE, kg m^−3^	NYE, kg kg^−1^	NPE, kg kg^−1^
Irrigation treatment			
FI	0.26 b *	5.94 b	26.98 b
0.75 FI	0.24 b	5.12 b	34.19 a
0.5 FI	0.31 a	7.87 a	31.55 ab
Nitrogen rate (kg N ha^−1^)			
125	0.31 a	7.90 a	29.77 a
250	0.28 a	6.64 a	34.08 a
375	o.22 b	4.40 b	28.87 a

* Means followed by the same letter in each trait are not significantly different at the 5% level of probability.

**Table 4 plants-11-02048-t004:** Soil and water properties.

Physical and Chemical Properties	Soil Depth (cm)			Irrigation Water
	0–30	30–60	60–90	
Field capacity (cm cm^−3^)	0.32	0.32	0.32	
Permanent wilting point (cm cm^−3^)	0.17	0.19	0.19	
Bulk density (g cm^−3^)	1.4	1.47	1.51	
Sand%	11	10	16	
Silt%	56	51	50	
Clay%	33	39	34	
Texture	SCL *	SCL	SCL	
EC (dS m^−1^)	0.74	0.51	0.49	0.58
Cl^−^ (meq L^−1^)	5.31	3.05	2.90	0.50
Na^+^ (meq L^−1^)	3.29	1.97	1.91	0.48
Ca^2+^ (meq L^−1^)	5.43	4.16	4.07	1.80
Mg^2+^ (meq L^−1^)	3.50	2.88	2.84	2.0

* Silty clay loam.

## Data Availability

The data presented in this study are available in the article.

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
