# Peer review of "Irrigation Regimes and Nitrogen Rates as the Contributing Factors in Quinoa Yield to Increase Water and Nitrogen Efficiencies"

_plants, 2022, doi:10.3390/plants11152048_

Round 1

Reviewer 1 Report

Line 20: was evaluated with a two-year field experiment.

Line 24: with.

Lines 25-28: Quinoa yield, WUE, and NUE of the treatment with 250kg N ha-1 and 0.75 FI should be described.

Lines 56-59: What do you mean with these sentences?

Lines 69-85: The interactive effects between nitrogen and irrigation should be mentioned.

Figure 1: It is hard to distinguish ET0 and0.75FI.

Line 114 and Line126: The 2.2 and 2.3 sections have same subtitle?

Lines 159-160: There was no two-way ANOVA results in Table 2. The experiment has two factors, i.e., irrigation and nitrogen, so the two-way ANOVA should be performed to evaluate the interactive effects of irrigation and nitrogen on quinoa yield and yield components.

Figure 2: The figure was distorted. High-resolution picture is needed.

Table 3: The two-way ANOVA should be performed to evaluate the interactive effects of irrigation and nitrogen on NUE, NYE, and NPE.

Lines 406-413: Seasonal rainfall in the two seasons should be present.

Line 416: Which nitrogen rate was adopted by local farmers?

Figure 6: Delete Chinese characters in X axis.

Line 443: The irrigation frequency is 7 days, right? What’s the irrigation rate?

Lines 449-451: Soil moisture only was measured in the two nitrogen treatments ( 0 and 250 kg N ha-1), how to determine soil moisture in other N treatments?

Lines 453-454: How to determine soil evaporation in other N treatments, as soil evaporation just measured in N1 and N3. The procedure for soil evaporation measurement should be described in detail.

Eqs. 5-7: which treatment was set was CK?

Author Response

Response to the reviewers’ comments                                                                           Date: July 26, 2022

Dear Prof. Eva Zeng,

The authors would like to sincerely thank you and respected reviewers for the accurate review and giving critical insight and extremely helpful comments. We have done our best to revise the manuscript according to all the comments and we hope that the revision is now acceptable to you and the reviewers for being published in Plants journal. We should note that all additions/modifications inside the text of the manuscript are presented in highlight. The point by-point responses to the reviewers' comments are also presented as follows.

Kind regards,

Didier Bazile and Rezvan Talebnejad, Corresponding authors

Reviewer 1#

Line 20: was evaluated with a two-year field experiment.

The relevant content is added as requested.

Line 24: with.

Thanks for this comment. The relevant content is corrected as requested.

Lines 25-28: Quinoa yield, WUE, and NUE of the treatment with 250kg N ha-1 and 0.75 FI should be described.

Thanks for this comment. The relevant content is added as requested.

Lines 56-59: What do you mean with these sentences?

Thanks for this comment. The content was presented to emphasize that deficit irrigation and high temperatures at the flowering and seed filling stages play an important role in quinoa seed yield.

Lines 69-85: The interactive effects between nitrogen and irrigation should be mentioned.

Thanks for this comment. The relevant content and literature review are added as requested.

Figure 1: It is hard to distinguish ET0 and 0.75FI.

Thanks for this comment. ETo is presented by a separated line with a smooth increasing trend as daily ETo was calculated. However, 0.75FI is presented with closed lines with a staircase trend to show individual irrigation events.

Line 114 and Line126: The 2.2 and 2.3 sections have same subtitle?

Thanks for this comment. 2.3 section is revised to seasonal evaporation and transpiration.

Lines 159-160: There was no two-way ANOVA results in Table 2. The experiment has two factors, i.e., irrigation and nitrogen, so the two-way ANOVA should be performed to evaluate the interactive effects of irrigation and nitrogen on quinoa yield and yield components.

Thanks for this critical comment. The interaction effects between irrigation regimes and N application rates were evaluated using two-way ANOVA. In case the interactive effects are significant the means are presented and compared in the structure like Table 2. However, in those cases where interactive effects are not significant, the main effect is presented (Table3).

Figure 2: The figure was distorted. High-resolution picture is needed.

Thanks for this comment. I try to improve the quality of the figure.

Table 3: The two-way ANOVA should be performed to evaluate the interactive effects of irrigation and nitrogen on NUE, NYE, and NPE.

Thanks for this comment. The interaction effects between irrigation regimes and N application rates were evaluated using two-way ANOVA. The interactive effects were not significant for NUE, NYE, and NPE. However, the main effect of irrigation regimes and the main effect of N application rates were significant. Therefore, the main effects are presented in Table 3.

 Lines 406-413: Seasonal rainfall in the two seasons should be present.

Thanks for this comment. The relevant content is added as requested. Rainfall depths during the two growing seasons were 29 and 64 mm, respectively.

Line 416: Which nitrogen rate was adopted by local farmers?

Thanks for this comment. N application rate by local farmers is 100-150 kg ha-1.

Figure 6: Delete Chinese characters in X axis.

Thanks for this comment. There are not any Chinese characters in the X axis. I guess the word software of the reviewer changes the language of the X axis.

Line 443: The irrigation frequency is 7 days, right? What’s the irrigation rate?

Thanks for this comment. Total irrigation depths were 850 and 714 mm for the two growing seasons, respectively.

Lines 449-451: Soil moisture only was measured in the two nitrogen treatments (0 and 250 kg N ha-1), how to determine soil moisture in other N treatments?

Thanks for this comment. Neutron probes were installed in the two nitrogen treatments because of the limitations of measuring devices. Therefore, the means of soil moisture were compared by Duncan’s multiple range test for these two N treatments at different irrigation regimes. (Table1.)

Lines 453-454: How to determine soil evaporation in other N treatments, as soil evaporation just measured in N1 and N3. The procedure for soil evaporation measurement should be described in detail.

Thanks for this comment. Micro-lysimeters were installed in the two nitrogen treatments because of the limitations of measuring devices. One small cylinder with 10 cm diameter and 30 cm height filled with the same field soil was buried in the surface soil as micro-lysimeters. The micro-lysimeters were weighted between irrigation intervals. Then, decreasing in micro-lysimeter weight was divided per the micro-lysimeter area to determine the evaporation from the soil surface.\

Eqs. 5-7: which treatment was set was CK?

Thanks for this comment. N application rate of 0 kg ha-1 was the control treatment.

Reviewer 2 Report

Dear authors,

please find the comments in a word document. 

Author Response

Reviewer 2#

Dear authors,

Congratulations on your demanding work and quality manuscript. Please find some suggestions here below. My recommendation is to improve material and method section regarding the weather conditions and irrigation scheduling.

Thanks for this general and useful comment. Total rainfall amounts and irrigation depths are added to the manuscript. Irrigation water depth was determined using the daily potential evapotranspiration and crop coefficient for quinoa with a 7-day interval.

L52 – kindly indicate why it is not mentioned that deficit irrigation is suitable for other climatic conditions?

Thanks for this comment. We want to emphasize the areas which suffer from water scarcity.

L54 – reference number is missing

Thanks for this comment. It was added to the text.

L56 – is this statement valid for comparison with rainfed treatments? What about the comparison with full irrigated crop?

Thanks for this question. Geerts et al. (2009) showed that deficit irrigation to 55% of full irrigation had no significant effect on quinoa dry matter compared with full irrigation treatments. However, the comparison with rainfed treatments is not evaluated in the article. In rainfed conditions spatial and temporal distribution of precipitation plays an important role in crop water requirement. Therefore, the amount of rainfall and its spatial and temporal distribution should be a consideration for comparison with deficit irrigation treatments.

L56 – 59 – these statements do not fit into the previous text

Thanks for this comment. The content was presented to emphasize that deficit irrigation and high temperatures at the flowering and seed filling stages play an important role in quinoa seed yield.

L66 – reference number is missing

Thanks for this comment. It was added to the text.

L74 – 85 – Is this valid for the topic of presented study?

Thanks for this question. Yes. The study area is located in arid and semi-arid weather conditions. Therefore, the descriptions is valid for study area.

L85 - Considering the previous paragraph and comment, state whether the aim of the work is to study treatments in non-native and water scarcity regions.

Thanks for this comment. It was added to the text.

Material and method section

Kindly elaborate Figure 6 in a way that you compare climate conditions (rainfall, air temperatures and air humidity) with long term averages.

Thanks for this comment. Because the quinoa growing season is about 4 months, therefore daily weather parameters are presented in the figure. However, with respect to the reviewer’s comment, the long-term average would not show the detailed weather condition during the growing season.

L409 – 413 - the rainfall analysis is missing

Thanks for this comment. The relevant content is added as requested. Rainfall depths during the wo growing seasons were 29 and 64 mm, respectively.

L427 – are the temperature values daily temperatures at planting date?

Thanks for this question. Yes. It was at sowing day.

Table 4. is not mentioned or elaborated in text

Thanks for this comment. It is mentioned at the end of the second paragraph in the materials and methods.

L442 - 443 – this statement is not clear enough.

Thanks for this comment. Total irrigation water depth was added to the text.

L449 – 451 – this statement is completely unclear

Thanks for this comment. Micro lysimeter was described by details in the text as requested.

Results

L91 – 92 – this is not clear enough, irrigation depths before irrigation

Thanks for this comment. Irrigation depths before applying irrigation regimes were presented. Irrigation treatments were not initiated at the planting date in order to have uniform germination and vegetative growth. Irrigation treatments were initiated at the vegetative with bud formation stage.

Figure 1 – kindly indicate the DAP aberration

Thanks for this comment. The relevant content is added as requested.

L100 – 101 – this sentence is completely unclear. How can irrigation have impact on soil water content before irrigation?

Thanks for this question. Deficit irrigation resulted in decrease in soil water content. The measurements of soil water content were conducted in specific time before each irrigation, when the soil water content reach its lowest level. Therefore, the average soil water content before irrigation events during crop growing season is presented in Table 1. for different treatments.

L103 – 104 – Usually, the soil nitrogen content is analyses in different irrigation treatments, yet you analyze the soil water content in different nitrogen treatment, which would be interesting if the soil organic matter, i.e. organic fertilizer was applied which is not your study goal

Thanks for sharing this interesting issue.

L102 – 103 – how is the significance determined (calculated)?

Thanks for this comment. The interaction effects between irrigation regimes and N application rates were evaluated using two-way analysis of variance (p<0.05).

L118 – 119 – what kind of statistical analysis did you performed for this statement?

Thanks for this comment. The interaction effects between irrigation regimes and N application rates were evaluated using two-way analysis of variance. (p<0.05). The relevant content is added.

L114- 126 – same subtitle

Thanks for this comment. 2.3 section is revised to seasonal evaporation and transpiration.

Kindly, explain why the evaporation is analyzed separately from ETo?

Thanks for this question. Separation of evaporation from evapotranspiration in experiments involved with N application rate treatments is important to distinguish the effect of canopy cover on the evaporation rate. Furthermore, deficit irrigation may reduce evaporation loss.

Discussion

L355 – Kindly indicate if the mention year was favorable or not unfavorable for quinoa growing due to the environmental conditions

 Thanks for this comment. The dramatic decrease in the seed yield at the first year occurred due to the late planting dates in the first year and air temperatures higher than 35ºC in the flowering stage. Therefore, first year sowing date was not favorable according to seed yield.

L382 – kindly, indicate the soil water content

Thanks for this comment. The soil water content is added as requested.

Reviewer 3 Report

Article interesting and very interesting

However, I have a few questions:

1. the abstract lacks information on the future of cultivation

2. how was irrigated, what was the chemical composition of the water used for irrigation?

3. what was used to control weeds (when, dose, what preparation?)?

4. the chemical composition of the soil in the years of the study ? why not given for the years?

5. why are the results given in terms of two years?

6. why are not the results given in terms of years of study?

7. why was there no interaction of nitrogen dose with irrigation?

8. please specify more precisely how the biometric traits were calculated?

9. the summary lacks the provision of agrotechnical recommendations

10. please correct the key words

Author Response

Reviewer 3#

Comments and Suggestions for Authors

Article interesting and very interesting

However, I have a few questions:

The abstract lacks information on the future of cultivation

Thanks for this comment. The relevant content is added at the end of the abstract as requested.

How was irrigated, what was the chemical composition of the water used for irrigation?

Thanks for this comment. The irrigation requirement is presented in 4.2. subsection. Irrigation water depth was determined using the daily potential evapotranspiration and crop coefficient for quinoa with a 7-day interval as surface basin irrigation. Irrigation water chemical water properties was added to Table 4.

What was used to control weeds (when, dose, what preparation?)

Thanks for this comment. Physical weed control was done every two weeks during the growing seasons. No herbicides were used.

The chemical composition of the soil in the years of the study? why not given for the years?

Thanks for this comment. The physical and chemical properties of the soil are presented in Table 4. These properties have no considerable variation during two years. Therefore, they are not measured for two years.

Why are the results given in terms of two years?

Thanks for this comment. Quinoa seed yield was significantly different between the two years (2017-2018) according to the analysis of variance (p < 0.05); therefore, the measured data was separately presented and discussed (Table 2).

Why are not the results given in terms of years of study?

Thanks for this comment. Quinoa total dry matter and other yield components were not significantly different in the two years (p < 0.05). Therefore, the mean values of these parameters over two years are presented in Table1., Table2 and Table3.

Why was there no interaction of nitrogen dose with irrigation?

Thanks for this comment. There is no significant interaction between the effect of irrigation regimes and nitrogen application rate on nitrogen use efficiencies according to the analysis of variance (p < 0.05).

Please specify more precisely how the biometric traits were calculated?

Thanks for this comment. Field measurements are presented in subsection 4.4. The relevant content is added as requested.

The summary lacks the provision of agrotechnical recommendations

Thanks for this comment. The following recommendations are presented at the end of the abstract.

Under non–limited water resource conditions, FI and N application rate of 375 kg ha-1 could be used for higher seed yields; however, under water-deficit regimes, N application rate of 250 kg ha-1 could be adequate. However, questions about which environmental factors impressively restricted the quinoa growth for optimizing the potential yield need further investigation.

Please correct the keywords

Thanks for this comment. Residual soil NO3-N was added to keywords.

Reviewer 4 Report

Generally, the paper has been well-conducted and written. Then, it is worth to be published in the journal. 

Just some minor points/doubts to consider:

Introduction: What's the meaning of non-native regions?

M&M: Include a picture (real conditions) with the plots 2x2 and the micro lysimeter.

Tabla 1: What happen with the N2 and N4 treatment? Just there are 2 treatments.

SWUE, DWUE? -- meaning?

2.4.1. Why was different the quinoa yield if the treatment receieved the same amount of water? Just this situation was due to higher air temperatures?

2.4.2. No references should appear in the Results sections. Please, move the paragraph to discussion

2.6. How do you measure residual soil NO3-N?

Equations 1,2, 3 Why you did no depict this equation in Figures? Is due to the number of samples? It would be better to present the information as Figures. 

Figure 4. In my opinion, it can be removed. 

Author Response

Reviewer 4#

Comments and Suggestions for Authors

Generally, the paper has been well-conducted and written. Then, it is worth being published in the journal. 

Just some minor points/doubts to consider:

Introduction: What's the meaning of non-native regions?

Thanks for this comment. Quinoa is an Andean pseudo-cereal that has been cultivated in the Andes area for at least 5000 years. Other areas are non-native regions for quinoa cultivation.

M&M: Include a picture (real conditions) with the plots 2x2 and the micro lysimeter.

Thanks for this comment. A picture was added to the manuscript in Figure 7.

Tabla 1: What happens with the N2 and N4 treatment? Just there are 2 treatments.

Thanks for this comment. Neutron probes were installed in the two nitrogen treatments because of the limitations of measuring devices. Therefore, the means of soil moisture were compared by Duncan’s multiple range test for these two N treatments at different irrigation regimes. (Table1.)

SWUE, DWUE? -- meaning?

Thanks for this comment. SWUE is seed water use efficiency and DWUE is dry matter water use efficiency. The relevant content is added to M.M as requested.

2.4.1. Why was different the quinoa yield if the treatment receieved the same amount of water? Just this situation was due to higher air temperatures?

Thanks for this comment. The dramatic decrease in the first year of seed yield occurred due to the late planting dates in the first year and air temperatures higher than 35ºC in the flowering stage (Fig. 6).

2.4.2. No references should appear in the Results sections. Please, move the paragraph to the discussion

Thanks for this comment. The comparison of quinoa transpiration efficiency was moved to the discussion section.

2.6. How do you measure residual soil NO3-N?

Thanks for this comment. Soil samples were taken from three depths 0-30 cm, 30-60 cm, and 60-90 cm, and air-dried for determining residual soil nitrate spectrophotometrically.

Equations 1,2, 3 Why you did no depict this equation in Figures? Is due to the number of samples? It would be better to present the information as Figures. 

Thanks for this comment. Equation 3. is presented in Figure 4 in the manuscript in accordance with the reviewer’s point of view. However, Equations 1. and 2. are simple linear relationships that are not presented as figures in the article in order to avoid extra figures. All the samples were available to drive the equation.

Figure 4. In my opinion, it can be removed. 

Thanks for this comment. With respect to the reviewer’s comment, it is useful to present Fig. 4 because it can be used easily by farmers or desiccation makers. The contour plots were developed in (Fig. 4) to show the combined effect of soil total available N and irrigation water depth on seed yield for practical use by farm managers

Round 2

Reviewer 1 Report

The manuscript has been improved substantialy. I suggeste that it can be accepted.

Reviewer 2 Report

Dear authors, thank you for the detailed answers and for making corrections to your manuscript. At this point, my only comment is regarding the long-term climate conditions. As it is stated in the first review round, the idea was to compare the climate data during the study period with the long-term data. That way, it is possible to state if the study period was dry, wet, or average in terms of weather conditions and therefore, favorable or not for quinoa growing. 

Congraturaltion and best wishes in future work

Reviewer 3 Report

Accepts corrections